# Integrated Analysis of the Metabolome and Transcriptome During Apple Ripening to Highlight Aroma Determinants in Ningqiu Apples

**DOI:** 10.3390/plants14081165

**Published:** 2025-04-09

**Authors:** Jun Ma, Guangzong Li, Yannan Chu, Haiying Yue, Zehua Xu, Jiaqi Wu, Xiaolong Li, Yonghua Jia

**Affiliations:** 1Horticultural Research Institute, Ningxia Academy of Agriculture and Forestry, Yinchuan 750002, China; 19995272146@163.com (J.M.); cynnx102030@163.com (Y.C.); yhyxxl@163.com (H.Y.); 13469684880@163.com (Z.X.); 18109539609@163.com (J.W.); 2Farmland Irrigation Research Institute, Chinese Academy of Agricultural Sciences, Xinxiang 453002, China; yzl025631@163.com

**Keywords:** apple, aroma, hub gene, apple tissue, ripening stage

## Abstract

We investigated the dynamic changes in volatile aroma compound profiles (types and concentrations) and associated gene expression patterns in both the peel and pulp tissues of apples during fruit maturation. This study aimed to elucidate the metabolic regulatory mechanisms underlying volatile aroma biosynthesis in Malus domestica “Ningqiu” apples, thereby providing theoretical support for the comprehensive utilization of aroma resources. Our methodological framework integrated headspace solid-phase microextraction gas chromatography–mass spectrometry (HS-SPME-GC-MS), ultra-high-performance liquid chromatography–orbitrap mass spectrometry (UHPLC-OE-MS), and Illumina high-throughput sequencing to generate comprehensive metabolomic and transcriptomic profiles of peel and pulp tissues. Critical differential aroma compound classes were identified, including esters, aldehydes, alcohols, terpenoids, and ketones, with their metabolic pathways systematically mapped through KEGG functional annotation. Our findings revealed substantial transcriptomic and metabolomic divergence across carotenoid, terpenoid, and fatty acid metabolic pathways. Integrative analysis of multi-omics data revealed 26 and 31 putative biologically significant hub genes in peel and pulp tissues, respectively, putatively associated with the observed metabolic signatures. Among these, five core genes—farnesyl diphosphate synthase (*FDPS1.X1*), alcohol acyltransferases (*AAT1* and *AAT3*), alcohol dehydrogenase (*ADH3*), and carotenoid cleavage dioxygenase (*CCD3*)—were recognized as shared regulatory determinants between both tissue types. Furthermore, terpene synthase (*TPS7*) emerged as a peel-specific regulatory factor, while hydroperoxide lyase (HPL2), alcohol dehydrogenases (*ADH2* and *ADH4*), and alcohol acyltransferase (*AAT2*) were identified as pulp-exclusive modulators of metabolic differentiation. The experimental findings provide foundational insights into the molecular basis of aroma profile variation in Malus domestica “Ningqiu” and establish a functional genomics framework for precision breeding initiatives targeting fruit quality optimization through transcriptional regulatory network manipulation.

## 1. Introduction

The biochemical characteristics of apples change significantly from harvest to ripening, with a double S-shaped growth curve including two rapid growth phases and an intermediate stagnation phase [1]. During the first growth phase, the fruit grows rapidly and accumulates a large number of metabolites that not only increase the acidity of the apple but are also precursors of volatile compounds, which are important for the flavor and aroma of the fruit. The beginning of the second growth stage, the turn-color stage (which marks the beginning of ripening), is a critical metabolic transition stage accompanied by a series of major changes, such as a gradual softening of the fruit, deepening of the color, a reduction in acidity, and an increase in sugar content. Many of the metabolites accumulated during the first stage remain in the fruit as it ripens, and their accumulation is usually tissue-specific, concentrated in the peel, pulp, or seeds [2].

Volatile organic compounds (VOCs) accumulated during fruit ripening are responsible for the unique aroma and flavor of fruits, including apples [3]. Volatile organic constituents are secondary metabolites, produced mainly by fatty acid and terpene metabolic pathways. Among them, aldehydes, alcohols, and esters are synthesized by the fatty acid metabolic pathway [4]. Firstly, glycerides are hydrolyzed by lipase to form free fatty acids, which serve as the initial substrates for subsequent reactions [5]. Subsequently, unsaturated fatty acids undergo oxidation, cleavage, and dehydrogenation reactions catalyzed by fatty acid desaturase (*FAD*), leading to the formation of aldehydes and alcohols. These reactions also involve the coordinated actions of lipoxygenase (*LOX*), hydroperoxide lyase (*HPL*) [6], and alcohol dehydrogenase (*ADH*) [7]. The resulting alcohols can be further catalyzed by alcohol acyltransferase (*AAT*) to form branched-chain esters, thereby enriching the aromatic profile. Additionally, terpene synthase (*TPS*) is a key enzyme in the synthesis of volatile terpenes [8]. Its substrates, isopentenyl diphosphate (*IPP*) and dimethylallyl diphosphate (*DMAPP*), are generated via the mevalonate (*MVA)* pathway in the cytoplasm and the methylerythritol phosphate (*MEP*) pathway in plastids. Under the catalysis of TPS, these precursors are converted into various terpenoid compounds, such as D-limonene, nerolidol, linalool, and β-ionone [9]. Meanwhile, carotenoids (e.g., α-carotene and β-carotene) can also be cleaved by carotenoid cleavage dioxygenase (*CCD*) to produce sesquiterpene ketones (e.g., α- and β-ionone) [10]. Although volatile compounds play a crucial role in fruit flavor, the mechanisms underlying their formation during apple ripening remain poorly understood.

Volatile compounds exist in both free and bound forms within the peel and pulp of apples [11]. Bound compounds are inherently odorless but can be converted into odor-active free compounds through hydrolysis during apple development [12]. Most volatile flavor components are primarily generated during the period from fruit color change to pre-harvest [13]. In previous comparative studies on varietal aroma, the “Ningqiu” variety (with a parentage of “Good Delicious” and “Red Astrachan”) garnered significant attention due to its rich flavor profile, making it highly valuable for research and worthy of further exploration. However, this variety has not been widely cultivated due to issues such as fruit drop susceptibility, soft pulp, poor storage and transport tolerance, and the perishability of ripe fruit. Nevertheless, with in-depth research into the synthesis and metabolic mechanisms of apple aroma, the research potential of the highly aromatic “Ningqiu” cultivar is expected to be fully realized. The breeding of apples as horticultural crops began with the utilization and improvement of local varieties by indigenous people [14]. Although these varieties hold significant value, information on their association with metabolites remains limited, and the core developmental transcriptome defined by aroma traits is primarily regulated by variety-specific gene expression [15]. In previous research, we conducted systematic profiling of volatile compound signatures and transcriptomic landscapes in Malus domestica “Qinguan” (QG) apples. Analytical characterization revealed that volatile ester biosynthesis constitutes the predominant metabolic signature in both peel and pulp tissues of mature QG fruits. Molecular validation through promoter transactivation assays confirmed that MdMYB94 transcriptionally activates the *MdAAT2* promoter, thereby establishing its functional role in modulating ester biosynthetic regulatory networks [16]. Comparative metabolomic profiling identified 35 volatile organic compounds (VOCs) across the Malus domestica “Granny Smith” and “Jonagold” cultivars. Aldehydes constituted the predominant class influencing aroma complexity in “Granny Smith” apples, whereas esters emerged as the principal volatile determinants in “Jonagold” specimens. Integrated pathway analysis revealed 94 differentially expressed genes (DEGs) functionally annotated to fatty acid metabolism, amino acid catabolism, mevalonate pathway, and phenylpropanoid biosynthesis regulatory networks [17]. However, due to the lack of transcriptomic and metabolomic data during the ripening process, as well as the focus on peel and pulp tissues rather than whole fruit samples, a comprehensive understanding of all changes occurring during apple ripening is yet to be achieved [18]. Through the joint analysis of metabonomics and transcriptomics (WGCNA), candidate genes were provided for the study of the molecular regulation mechanism of fruit quality, in order to clarify the characteristics of aroma components in the tissue of “Ningqiu” apple fruit in this region and to preliminarily analyze the molecular regulation mechanism of key differential aroma substances.

In this study, we investigated the metabolic dynamics of flavor during fruit ripening by detecting changes in firmness, soluble solid content (SSC), and VOCs during the ripening stages of “Ningqiu” apple fruits, i.e., the green fruit stage, 30 d after anthesis (DAH), the color-transferring stage, 60 DAH after anthesis, and the fruit ripening stage, 90 DAH after anthesis. Through comprehensive analysis of metabolites and transcriptomes related to peel and pulp flavor, we constructed a regulatory network controlling the production and accumulation of volatiles. Importantly, we also summarized the key transcription factors (TFs) that regulate VOC metabolism in different fruit species. These results help to elucidate the molecular mechanisms of flavor and further promote the study of regulatory mechanisms for fruit quality improvement.

## 2. Materials and Methods

### 2.1. Materials

Samples of “Ningqiu” apples (“Good Delicious” × “Red Astrachan”) were obtained from the National Comprehensive Experimental Base for Apples, located at the Institute of Horticulture, Ningxia Academy of Agricultural and Forestry Sciences in Ningxia, China (38°38′ N, 106°09′ E). This region experiences a temperate continental climate, with an annual rainfall of 200–220 mm and an average temperature ranging from 5 °C to 18 °C. Approximately 5 kg each of apples were randomly harvested from three trees selected for their uniform fruit weight, tree structure, and growth conditions. All collected apples were free from visible defects such as rot, disease, or insect damage. To ensure uniformity in ripening stages, the starch–iodine index was assessed using the method described by Blanpied and Silsby [19], categorizing the fruits into three stages: green fruit (S1), color-turning (S2), and ripening (S3). Immediately after collection, the skin and pulp of the apples were separated, flash-frozen in liquid nitrogen, and stored at −80 °C for subsequent analysis.

The phenotypic characteristics of “Red Astrachan” were as follows: apple weight (S1: 142 g; S2: 158 g; S3: 208 g); TSS (S1: 3.66%; S2: 8.29%; S3: 12.1%); apple width (S1: 40 mm; S2: 60 mm; S3: 74 mm); and apple length (S1: 34 mm; S2: 53 mm; S3: 61 mm). The phenotypic characteristics of “Golden Delicious” were as follows: apple weight (g) (S1: 143 g; S2: 178 g; S3: 192 g); TSS (S1: 4.32%; S2: 9.62%; S3: 13.9%); apple width (S1: 35 mm; S2: 57 mm; S3: 71 mm); and apple length (S1: 31 mm; S2: 54 mm; S3: 60 mm).

### 2.2. Fruit Quality

The soluble solid content (TSS) of the fruits was measured using a portable refractometer (Model PAL-2, ATAGO, Tokyo, Japan). Fruit firmness was assessed using a texture analyzer (TA-XT Plus, Stable Micro Systems, Godalming, UK). The titratable acidity (TA) was quantified through acid–base titration, as described in [20]. For each measurement, a sample of five fruits were analyzed, and the experiment was repeated three times to obtain the mean value.

The “Ningqiu” apple is a red variety selected from a cross between “Good Delicious” and “Red Astrachan” (Figure 1). Fruit firmness, soluble solids, and titratable acidity are key factors in fruit quality, with soluble solids showing a significant increase during fruit development and TSS, mainly sugars in the fruit, accumulating with ripeness for better flavor and aroma (Figure 1A). Fruit firmness and titratable acidity showed significant decreases throughout ripening, with green fruit stage (S1) fruit usually being crisp but lacking in flavor and aroma, and with decreasing firmness, flavor and aroma may be optimal; meanwhile, organic acid content can mask sweetness, affecting the flavor balance (Figure 1B,C). TSS and TA together determine the flavor and aroma of the fruit, and the appropriate firmness and sugar–acid ratio can facilitate the best taste and flavor experience.

### 2.3. Determination of Aroma Components

Volatile compounds were extracted from apple samples using headspace solid-phase microextraction (HS-SPME), following a previously established method [21]. Frozen apple peel or pulp stored at −80 °C was ground into a fine powder using liquid nitrogen and a mortar and pestle. Approximately 3 g of the powdered sample was transferred into a 20 mL headspace vial, and 3 mL of saturated NaCl solution was added for analysis. The extracted volatiles were analyzed using a gas chromatography system (TRACE 1310, Thermo Scientific, Waltham, MA, USA) coupled with a triple quadrupole mass spectrometer (TSQ 9000, Thermo Scientific, Waltham, MA, USA). Samples were injected in split mode (1:5 ratio), with approximately 17% of the injected volume directed to a non-polar column (TG-5 MS, 30 m × 0.25 mm ID, 0.25 µm film thickness, Thermo Scientific, Waltham, MA, USA) using helium as the carrier gas at a flow rate of 1.0 mL/min. The mass spectrometer operated in electron ionization (EI) mode at 70 eV, with an ion source temperature of 230 °C, a scan rate of 2.88 scans/s, and a mass range of *m/z* 29–540. Compounds were tentatively identified by matching their mass spectra with the NIST mass spectral library. Quantification was performed based on total ion chromatogram (TIC) peak areas, using 2-octanol as an internal standard for normalization.

### 2.4. Transcriptome Analysis

#### 2.4.1. RNA Extraction, Library Construction and Quality Control, and Sequencing

Referring to previous methods [22,23], the quality of total RNA was assessed using a NanoDrop 2000 spectrophotometer (Thermo Scientific, Waltham, MA, USA) to determine concentration and purity, while integrity was evaluated via RNA-specific agarose electrophoresis or the Agilent 2100 Bioanalyzer (Agilent Technologies, Santa Clara, CA, USA) with the RNA 6000 Nano Kit (Cat. No. 5067-1511). Total RNA samples with a quantity of ≥ 1 µg were selected for library preparation using the NEBNext Ultra II RNA Library Prep Kit for Illumina (New England Biolabs, Ipswich, MA, USA), a strand-specific library construction kit.

The NEBNext Ultra Directional RNA Library Prep Kit for Illumina was used to construct the RNA library. mRNA with polyA tails were enriched using Oligo(dT) magnetic beads and then fragmented into short strands using divalent cations under elevated temperature. The fragmented mRNA served as a template for first-strand cDNA synthesis using random oligonucleotide primers, followed by second-strand cDNA synthesis. The double-stranded cDNA was purified, subjected to end repair, and ligated with sequencing adaptors after adding an “A” base to the 3’ ends. The cDNA fragments were size-selected using AMPure XP beads (Beckman Coulter, Brea, CA, USA), amplified by PCR, and further purified with AMPure XP beads.

Library quality was assessed using the Agilent 2100 Bioanalyzer (Agilent Technologies, Santa Clara, CA, USA) and the Agilent High Sensitivity DNA Kit (Cat. No. 5067-4626). Total library concentration was quantified using the Quant-iT PicoGreen dsDNA Assay Kit (Invitrogen, Waltham, MA, USA, Cat. No. P7589) and a Quantifluor-ST fluorometer (Promega, Madison, WI, USA, Cat. No. E6090). Validated library concentrations were further confirmed by qPCR (StepOnePlus Real-Time PCR Systems, Thermo Scientific, Waltham, MA, USA). Equimolar amounts of multiplexed DNA libraries were pooled, diluted, and sequenced in PE150 mode on an Illumina platform (Illumina Inc., San Diego, CA, USA).

#### 2.4.2. Analysis of Participant Transcriptome Data

Referring to previous methods [24,25,26,27,28,29,30], the following processes were performed.

Data Quality Control: Following sequencing, raw image files were processed by the sequencing platform’s software to generate FASTQ-formatted raw data. To ensure data reliability, low-quality reads and adapter sequences were filtered out using fastp (v0.22.0). The filtering criteria included (1) the removal of adapter sequences at the 3’ end and (2) the exclusion of reads with an average quality score below Q20. All subsequent analyses were performed using high-quality clean data.

Sequence Alignment with the Reference Genome: The reference genome and corresponding gene annotation files were downloaded from a public genome database. The genome index was constructed using HISAT2 (v2.1.0), and paired-end clean reads were aligned with the reference genome. HISAT2 was selected for its ability to generate a splice-aware alignment database based on gene annotation files, providing superior alignment accuracy compared to non-splice-aware tools.

Gene Expression Analysis: Gene expression levels were quantified using HTSeq (v0.9.1) to calculate the read count values for each gene. To enable cross-sample and cross-gene comparisons, expression levels were normalized using FPKM (fragments per kilobase of transcript per million mapped fragments). For paired-end sequencing, FPKM only counts fragments where both reads align with the same transcript.

Differential Expression Analysis: Differential expression analysis between sample groups was performed using DESeq (v1.38.3). Differentially expressed genes (DEGs) were identified based on the following thresholds: |log2FoldChange| > 1 and *p*-value < 0.05.

Enrichment Analysis: Gene Ontology (GO) enrichment analysis was conducted using topGO (v2.50.0). The hypergeometric distribution method was applied to calculate *p*-values, with significant enrichment defined as *p*-value < 0.05. This analysis identified GO terms significantly enriched with differentially expressed genes (all, up-regulated, or down-regulated), providing insights into their primary biological functions. Kyoto Encyclopedia of Genes and Genomes (KEGG) pathway enrichment analysis was performed using clusterProfiler (v4.6.0), focusing on pathways with significant enrichment (*p*-value < 0.05). Additionally, Gene Set Enrichment Analysis (GSEA, v4.1.0) was employed to evaluate predefined gene sets without requiring a predefined differential expression threshold. All genes were ranked based on their differential expression levels between sample groups, and statistical tests were conducted to determine whether predefined gene sets were significantly enriched at the top or bottom of the ranked list.

### 2.5. Untargeted Metabolomics (UHPLC-OE-MS) Analysis

#### 2.5.1. Metabolite Extraction

Approximately 25 mg of each sample was weighed out into a pre-chilled EP tube, followed by the addition of homogenization beads and 500 µL of the extraction solvent (methanol/acetonitrile/water = 2:2:1, *v*/*v*) containing isotopically labeled internal standards. The mixture was vortexed for 30 s and homogenized using a homogenizer (35 Hz, 4 min). Subsequently, the samples were sonicated in an ice-water bath for 5 min, and this step was repeated three times. After homogenization, the samples were incubated at −40 °C for 1 h and then centrifuged at 4 °C for 15 min at 12,000 rpm (centrifugal force: 13,800× *g*; radius: 8.6 cm). The supernatant was collected and transferred into an injection vial for analysis. Additionally, equal volumes of supernatant from all samples were pooled to create quality control (QC) samples, which were also analyzed [31].

#### 2.5.2. On-Board Testing

Non-polar metabolites were analyzed using a Vanquish UHPLC system (Thermo Fisher Scientific, Waltham, MA, USA) coupled with a Phenomenex Kinetex C18 column (2.1 mm × 50 mm, 2.6 µm) for chromatographic separation. The mobile phase consisted of solvent A (0.01% acetic acid in water) and solvent B (isopropanol/acetonitrile = 1:1, *v*/*v*). The sample tray was maintained at 4 °C, and the injection volume was set to 2 µL. Mass spectrometry data were acquired using an Orbitrap Exploris 120 mass spectrometer controlled by Xcalibur software (v4.4, Thermo). The instrument parameters were as follows: sheath gas flow rate, 50 Arb; auxiliary gas flow rate, 15 Arb; capillary temperature, 320 °C; full MS resolution, 60,000; MS/MS resolution, 15,000; normalized collision energy (NCE), 20/30/40; and spray voltage, 3.8 kV (positive mode) or −3.4 kV (negative mode) [32,33].

#### 2.5.3. Metabolomic Data Analysis

Referring to previous methods [34,35,36], raw data parameters, including retention time and mass-to-charge ratio (*m/z*), were initially screened using CD(24.3.0.571) search software. Peak alignment across samples was performed with a retention time deviation threshold of 0.2 min and a mass deviation of 5 ppm to ensure accurate identification. Peak extraction was conducted based on predefined criteria: mass deviation ≤ 5 ppm, signal intensity deviation ≤ 30%, signal-to-noise ratio ≥ 3, minimum signal intensity ≥ 100,000, and consideration of adduct ions. Peak areas were quantified, and target ions were integrated for further analysis.

Molecular formulas were predicted using molecular ion peaks and fragment ions, and the results were cross-referenced with the mzCloud (Date of visit: 11 October 2024, https://www.mzcloud.org/), mzVault(Compound Discoverer 3.3), and Masslist (NIST Chemistry WebBook and METLIN) databases. Background ions were removed via comparison with blank samples, and quantitative results were normalized. The final output included both the identification and quantification of metabolites.

Unsupervised principal component analysis (PCA) was employed to assess inter-group differences and intra-group variability among samples, with the correlation between quality control (QC) samples visualized using a heat map. To identify significantly altered metabolites, variable importance in projection (VIP) values were derived from a combination of Student’s *t*-test and supervised partial least squares–discriminant analysis (PLS-DA). The PLS-DA model was validated through 200 permutation tests to prevent overfitting. Differentially accumulated metabolites (DAMs) were screened based on the following criteria: |Log2FC| > 1.5, *p*-value < 0.05, and VIP > 1. These metabolites were subsequently subjected to KEGG pathway enrichment analysis to identify key metabolic pathways associated with the observed differences.

### 2.6. Unsigned Weighted Correlation Network Analysis (WGCNA)

#### 2.6.1. WGCNA for Hierarchical Clustering and Identification of Co-Expressed Genes (“Hub Genes”)

These genes are likely to play a pivotal regulatory role and/or significantly influence phenotypic traits [37]. Specifically, two distinct adjacency matrices were constructed using FPKM expression values from the peel and pulp tissues of Ningqiu apples at three developmental stages (S1, S2, and S3). The optimal soft threshold power was determined using the pickSoftThreshold function, with the lowest power achieving a scale-free topology fit index of 0.90 being selected for each analysis [38,39]. The weighted adjacency matrix was subsequently transformed into a topological overlap matrix (TOM) to reduce spurious associations, and dissimilarity values derived from TOM were used for hierarchical clustering. Highly interconnected gene clusters, referred to as modules, were identified in the resulting dendrogram using the Dynamic Hybrid TreeCut algorithm. Finally, the correlation between each module and metabolite levels was evaluated by calculating Pearson correlation coefficients based on module eigengene values, as described by Esposito et al. [40] and Vernocchi et al. [41].

#### 2.6.2. Hub Gene Analysis

Hub genes are defined based on module affiliation (MM) and gene significance (GS) values, both calculated by WGCNA. The former is defined as the correlation between the gene expression profile and the module characterized genes (ME), thus explaining the proximity of a gene to a given module [42]. The latter measure defines the absolute value of the correlation between individual genes and metabolite accumulation. Module hub genes were selected according to GS > 0.2 and MM > 0.8 (*p*-value < 0.05) based on external traits [42].

### 2.7. RT-qPCR

For cDNA synthesis, random hexamers were utilized to anneal with DNase-treated RNA, and reverse transcription was carried out using the PrimeScriptTM RT kit at 37 °C for 30 s. The synthesized cDNA was diluted with ddH2O at a 1:3 ratio and stored at −20 °C. RT-qPCR was performed on the Bio-Rad CFX96 platform. Gene-specific primers (Appendix A), designed using Primer Premier 5 software, were used for amplification, with ACTIN as the internal reference gene. Each sample was tested in three technical replicates, and the relative expression levels of target genes were determined by averaging the results of the replicates. Normalization was performed using the 2^−ΔΔCt^ method, based on the Ct values of the internal control gene [43].

### 2.8. Statistical Analysis of Data

The raw data were first converted to mzXML format using ProteoWizard 3.0.9134. Metabolite identification was subsequently conducted with a collaboratively developed R package, based on the BiotreeDB (V3.0) database [44]. Data visualization and analysis were then performed using an independently written R 4.4.1 package, alongside the R language, Origin 2021, and TBtools-II software.

## 3. Results

### 3.1. Overview of VOC Accumulation Patterns During Fruit Ripening

We analyzed the VOCs using GC-MS, and the total ion mobility plot baseline was stable, while the corresponding peak intensities and retention times largely overlapped, indicating that the experimental assay results were reliable (Appendix A). The relative changes in VOCs and content of the “Ningqiu” variety were determined during fruit development. In total, 24 and 170 volatiles were measured in the peel and pulp tissues, respectively. Of these, 19 alcohols, 49 esters, 23 aldehydes, 10 ketones, 25 terpenes, 9 acids, 32 alkanes, and 36 others were found in the peel tissue (Appendix A), while 15 alcohols, 35 esters, 18 aldehydes, 10 ketones, 7 terpenes, 15 acids, 24 alkanes, and 45 others were found in the pulp tissue (Appendix A). The proportion of esters and alcohols gradually increased with ripening, and aldehydes showed higher levels before color change and decreased after color change (Figure 2A,B). Hierarchical clustering of VOC spectra showed that the 107 volatiles from the peel and pulp could be generally classified into three sub-groups (Appendix A). Sub-group 1 contained 76 VOCs such as esters, alkanes, terpenes, and olefins, which were highly accumulated mainly in the peel at the mid-ripening (S3) stage. Sub-group 2 consisted of VOCs such as terpenoids, olefins, and esters, which were highly expressed at the varietal color change (S2) stage in the pulp, showing significant varietal and temporal specificity. The VOCs in sub-group 3, which mainly consisted of aldehydes, alkanes, alcohols and terpenoids, showed high accumulation at the green fruit (S1) stage. Considering that alkanes have low aroma activity as well as minimal acid content, differences in ester, terpene, aldehyde, and alcohol content between peel and pulp tissues were responsible for the differences in fruit odor.

### 3.2. Integration-Associated DEGs and DMAs Provide Molecular Insights into Flavor Formation

In transcriptomics, significant changes in the expression of DEGs were observed in the peel and pulp tissues at the three stages, with *p*-value < 0.05 denoting the case of significant enrichment of differentially expressed genes. In the peel comparator group, in the differentially expressed genes SKS2 vs. SKS1, 1198 were up-regulated and 1891 were down-regulated, while in SKS3 vs. SKS1, 2945 were up-regulated and 9137 were down-regulated. In the pulp, in FPS2 vs. SKS1, 1555 were up-regulated and 1533 were down-regulated, while in FPS3 vs. SKS1, 3188 were up-regulated and 7804 were down-regulated) (Appendix A). To understand the metabolic pathways and biological functions involved in the differentially expressed genes, we performed KEGG enrichment analysis of these differential genes (Appendix A).

In the metabolomics counterpart, we used UHPLC-OE-MS for untargeted metabolomics analysis. Comparison and correlation analyses of the total ion flow map spectra of quality control (QC) samples for a total of 13 samples from three periods showed that the baseline of the total ion flow map was stable in both positive and negative ion modes, and the corresponding peak intensities and retention times largely overlapped (Appendix A), with a correlation coefficient of greater than 0.9 among the QC samples (Appendix A). The results of the PCA revealed that the parallel sample groups of peel (SKS1, SKS2, and SKS3) and pulp (FPS1, FPS2, and FPS3) were clustered together, suggesting that the samples were selected reasonably well and that metabolites differed between sample groups (Appendix A). We compared the four developmental stages between groups and used OPLS-DA to screen for differential metabolites, and the longitudinal intercept of the Q2 regression line for each comparison group was on the negative half axis of the *y*-axis (Appendix A). Indicating the validity of the PLS-DA model validation, we used an FC > 2, a VIP > 1, and a *p*-value < 0.05 as the screening conditions, and there were significant differences between peel and pulp metabolites at the three developmental stages among the peel comparison groups in SKS2 vs. SKS1 (426 up-regulated and 943 down-regulated), in SKS3 vs. SKS1 (2938 up-regulated and 2308 down-regulated), in (744 up-regulated, 508 down-regulated), in FPS2 vs. FPS1 (743 up-regulated, 508 down-regulated), and in FPS3 vs. FPS1(1916 up-regulated, 926 down-regulated) in the fruit pulp (Appendix A). To understand the metabolic pathways and biological functions involved in the differential metabolites, we performed KEGG enrichment analysis of these differential metabolites (Appendix A).

We identified the pathways of terpenoid backbone biosynthesis, carotenoid biosynthesis, and those related to esters, terpenoids, aldehydes, and alcohols through the metabolic pathways of differentially expressed gene KEGG enrichment (Appendix A) and metabolite KEGG enrichment (Appendix A), including linoleic acid metabolism and alpha-linolenic acid metabolism. In subsequent sections, we summarize the transcriptomic and metabolomic patterns of DEGs and DAMs involved in the biosynthesis of apple aroma compounds and their precursors through the green fruiting (S1) stage of apple aroma compounds and their precursors versus the transcoloration (S2) and ripening of the epidermis and pulp (S3). We produced provisional customized metabolic profiles for each pathway, including gene expression and chemical data.

#### 3.2.1. Terpene Synthesis Pathway Expression Patterns

Transcriptomic analyses were able to identify DEGs involved in the biosynthesis of plastidic MEV and cytoplasmic MVA terpenes. MVA begins with the condensation of acetyl coenzyme A. In a reaction catalyzed by acetyl coenzyme A C-acetyltransferase (*ACAT*), two acetyl coenzyme As are condensed to form acetyl coenzyme A. Acetyl coenzyme A and acetyl coenzyme A are then condensed by 3-hydroxy-3-methylglutaryl coenzyme A synthase (HMGS) to form 3-hydroxy-3-methylglutaryl coenzyme A (HMG-CoA). HMG-CoA is condensed in the presence of the enzyme HMGS. HMG-CoA is reduced to mevalonate in a reaction catalyzed by 3-hydroxy-3-methylglutaryl coenzyme A reductase (HMGR). The identified 1-deoxy-D-xylulose-5-phosphate synthase *DXS2* (LOC103401537), *DXS3* (LOC103453010), and *DXS4* (LOC103415150) genes were induced in the peel in different fruit tissues and were active after the green fruit stage. DXS1 (LOC103403800) was repressed in the pulp peel after the green fruit stage, while DXS4 showed progressively higher transcript levels in the peel throughout ripening (Figure 3 and Appendix A). Geranyl pyrophosphate synthase (GPPS) is a key downstream gene in the terpene pathway, capable of converting isopentenyl pyrophosphate (IPP) or dimethylallyl pyrophosphate (DMAPP) to geranyl pyrophosphate (GPP), a precursor for terpene synthesis.

Downstream of the MEP and MVA pathways, terpenoids are mainly regulated by terpene synthases (*TPSs*), which are encoded by a large class of enzymes. Notably, 11 (in the peel) and 7 (in the pulp) genes encoding *TPS* were differentially expressed. In the peel, 11 TPSs were activated during ripening, of which only *TPS1* (LOC103417108) was highly expressed at the green fruit stage (S1), while *TPS3* (LOC103448570), TPS4 (LOC114819100), *TPS5* (LOC103445375), and *TPS11* (LOC103410286) showed medium–high transcript levels at the color change (S2) and ripening (S3) stages (Figure 3 and Appendix A). *TPS1, TPS5, TPS7*(LOC103435869), *TPS8* (LOC103430135), and *TPS11* were repressed in the analyzed pulp tissues during ripening (Figure 3 and Appendix A). Farnesyl diphosphate synthase (*FDPS*) is a key enzyme in the biosynthesis of terpenoids in apples and catalyzes the generation of FDP from IPP and DMAPP. *FDP* is a direct precursor of sesquiterpenoids (e.g., germacrone and artemisinin) and highly accumulates in the peel.

#### 3.2.2. Changes in the Expression of Volatile-Compound-Related Genes Involved in Fatty Acid Metabolic Pathways

Linoleic and linolenic acids are precursors for the synthesis of branched-chain fatty alcohols, aldehydes, ketones, and esters. Linoleic acid can be converted to linolenic acid by the enzyme fatty acid desaturase (*FAD*). The *LOX* pathway uses linolenic acid as a precursor for the synthesis of related VOCs, which is regulated by a series of genes. With regard to the transcriptional profile of the fatty acid metabolic pathway, five key genes, namely genes encoding for fatty acid desaturases (*FADs*), lipoxygenases (*LOXs*), hydroperoxide lyases (*HPLs*), alcohol dehydrogenases (*ADHs*), and alcohol acetyltransferases (*AATs*), were considered. Three *FADs* were identified in the peel and pulp, and their expression patterns varied according to developmental stage. *FAD1* (LOC103443127) was highly expressed at the color change (S2) and ripening (S3) stages; FAD2 (LOC103447141) was highly expressed only in the peel and was most active at the ripening (S3) stage in the different tissues of the fruits studied; and *FAD3* (LOC103407115) was repressed in the pulp and up-regulated in the peel at the green stage (Figure 4 and Appendix A).

Lipoxygenase (*LOX*) and hydroperoxide lyase (*HPL*) are key enzymes in aldehyde biosynthesis, and the five *LOX* DEGs and two HPLs that we identified in fruit tissues were basically highly expressed at the green fruit stage (S1), while *LOX3* (LOC103447063) and *HPL2* (LOC114819168) were inhibited throughout the reproductive period (in the pulp). It is worth noting that *LOX* and *HPL* were down-regulated in the peel after the green fruit stage (S1), whereas they were down-regulated in the peel from the color change stage onward, suggesting that the accumulation of aldehyde precursors in the peel was more prolonged compared to that in the pulp, and that this might be related to the presence of heptanal, (E)-2-heptenal, benzeneacetaldehyde, (E)-2-octenal, nonanal, decanal, (E,E)-2-octenal, and (E)-2-octenal in the peel. (E)-2,4-nonadienal is associated with high accumulation in the green fruit stage (S1). Aldehydes can be further converted to alcohols by alcohol dehydrogenase (*ADH*), and both *ADH1* (LOC103409569) and *ADH3* (LOC103439766) were highly expressed in fruit tissues at the ripening stage (S3), which may be related to 1-butanol and 1-hexanol; in addition, compared to the pulp, *ADH2* (LOC103432769), *ADH4* (LOC114827344), *ADH5* (LOC103435799), *ADH6* (LOC114819103), and *ADH7* (LOC103428551) were found at higher expression levels in peel transcripts prior to ripening (S3), while compounds with a high content of alcohol metabolites were also concentrated in the peel.

Each transcript of the *AAT* enzyme function, which plays a key role in ester synthesis, behaved differently and showed highly variable expression patterns and values in fruit tissues (Figure 4 and Appendix A). We identified four *AAT* DEGs in fruit tissues (Figure 4 and Appendix A). Interestingly, in the peel and pulp, most of their DEGs and ester DAMs were highly expressed during the color change stage (Figure 4 and Appendix A).

#### 3.2.3. Changes in the Expression of Genes Related to Volatile Compounds in the Terpenoid and Carotenoid Pathways

In order to study the behavior of carotenoids and carotenoid-pathway-related aroma genes, 38 genes presumed to be involved in their catabolism were selected [15]. More specifically, they showed different expression patterns during ripening, confirming the complexity of gene regulation of the pathway. For example, 11 genes were highly expressed throughout ripening, with some (e.g., carotenoid 9,10(9′,10′)-cleavage dioxygenase 1 *CCD2* (LOC103413126), beta-carotene 3-hydroxylase *CrtZ2* (LOC103439793), *CrtZ3* (LOC114825667), beta-ring hydroxylase *LUT5.2* (LOC103427772), and (+)-abscisic acid 8’-hydroxylase *CYP707A1.4* (LOC103455974)) only in the skin and others (e.g., 9-cis-epoxycarotenoid dioxygenase *NCED2* (LOC103431063)) only in the pulp (Figure 5 and Appendix A).

GC-MS and LC-HRMS analyses of four carotenoids and four carotenoid precursors, respectively, highlighted the different prevalence of this group of compounds in the two organizations. It is well known that damascenone and geranyl acetone, as carotenoid cleavage derivatives, tend to accumulate in fruits with various tendencies during ripening. Previous studies reported higher accumulation in Baga and Pinot Noir during ripening. Our results confirmed this finding in the peel but not in the pulp (Figure 5 and Appendix A). Fruit pulp showed similar trends in all groups, except that the difference in the trend of lower changes in transcript levels of DAMs found at ripening was not significant (overaccumulation of two compounds and decline of two compounds). Regarding carotenoid precursors, they were strongly influenced by variety and organization. Throughout ripening, zeaxanthin and zeaxanthin were highly accumulated in the skin and decreased in the pulp (Figure 5 and Appendix A).

### 3.3. WGCNA

WGCNA identified 32 and 31 different highly co-expressed modules (gene clusters) in the peel and pulp, respectively (Figure 6A,B). The largest module (yellow–green for the peel and green for the pulp) consists of 8314 and 7360 genes, while the smallest module (coral1 for the peel and skyblue2 for Falanghina) contains 98 and 79 genes, respectively (Figure 6A,B). In this proof-of-concept study, we delved deeper into modules containing at least one previously described aroma gene (19 modules in the peel (Figure 6C, Appendix A) and 20 modules in the pulp (Figure 6D, Appendix A)). A significant correlation does not necessarily imply a causal relationship between genes and metabolites; however, it allows us to propose potential candidates for gene function and discard genes unrelated to metabolites. As for the fruit peel, the green–yellow module contains 30 aroma genes, belonging to terpenes (11), lipoxygenases (12), and carotenoids (7). This module is related to lutein, hexanal, octanal, benzeneacetaldehyde, tridecanal, dibutyl phthalate, abscisic acid, cyclofenchene, (E)-β-farnesene, α-longipinene, δ-guaiene, cis-bisabolene, turmeronol B, germacrone, 1-hexanol, (Z)-2-octen-1-ol, acetic acid, butyl ester, butyl 2-methylbutanoate, and alpha-linolenic acid (Figure 6C). The most representative module in fruit pulp is the green module, which contains 19 aroma genes belonging to terpenes (9), lipoxygenases (6), and carotenoids (4). This module is related to damascenone, zeaxanthin, heptanal, benzeneacetaldehyde, dimethyl-silanediol, cis-3-hexenyl iso-butyrate, trans-beta-ionone, turmeronol B, hexanal, (E)-2-octenal, and dibutyl phthalate (Figure 6D).

In selected modules, we focused on hub genes (genes with the highest intramodal connectivity), which may represent biological points of interest that define specific metabolic patterns (Appendix A). The overall correlation values between metabolite levels and hub gene expression were higher in the peel than in the pulp. In particular, 26 hub genes were selected in the 11 peel modules (Appendix A), whereas 31 hub genes were selected in the 12 pulp modules (Appendix A). Among the peel hub genes, ten were terpene-related genes; nine were lipoxygenase genes, of which four *ADH* genes were positively correlated with various metabolites; and seven were key carotenoid genes (Appendix A).

Among the differentially expressed fruit peel hub genes, several are associated with metabolites belonging to the same pathway they participate in. For example, in the terpene pathway, *TPS* (LOC103435869) is positively correlated with geranial (0.892), santolina triene (0.892), (E, E)-α- farnesene (0.881), and (−)-β-chamigene (0.835) in the dark multi-agent module. *FPDS* (LOC103445233) is positively correlated with germatrone (0.931) and turmeronol B (0.902) in the maroon module. In the lipoxygenase pathway, *ADH* (LOC103432769) is positively correlated with 1-butanol (0.957) and 1-hexanol (0.927) in the magenta module. *AAT* (LOC1034367900) and *AAT* (LOC103447337) are used in conjunction with the dark magenta module, which includes butylic acid hexyl ester (0.895), butylic acid butyl ester (0.892), L-alanine methyl ester (0.892), 2-methyl butylic acid butyl ester (0.892), butylic acid pentyl ester (0.892), 2-methyl-butanoic acid heptyl ester (0.885), 2-methyl-propanoic acid hexyl ester (0.884), propanoic acid hexyl ester (0.868), and 2-methyl-butanoic acid. Acid hexyl ester (0.860) is positively correlated. In the carotenoid pathway, *CCD3* (LOC103438403) and trans-beta-ionone (0.879) are positively correlated in the Thistle2 module (Appendix A).

In the pulp, the identified hub genes belong to the carotenoid (12), lipoxygenase (11), and terpene (8) pathways (Appendix A). In the terpene pathway, *FPDS* (LOC103445233) is positively correlated with germatrone (0.898), turmeronol B (0.887), and amidepine (0.885) in the Greenhill module. In the lipoxygenase pathway, *HPL* (LOC114819168) is positively correlated with (E)-2-Heptenal (0.885) in the Greenhill module. *ADH* (LOC103439766) is positively correlated with (E)-2-octenal (0.921) and hexanal (0.918) in the coral1 module. *ADH* (LOC103418393) is positively correlated with nonanal (0.924), decanal (0.924), and octanal (0.852) in the dark red module. *ADH* (LOC114827344) is positively correlated with (E)-2-octenal (0.839) and hexanal (0.832) in the dark turquoise module. *ADH* (LOC103432769) is positively correlated with β-cyclocitral (0.977), (E)-2-heptenal (0.977), and 2,4-decadienal (0.877) in the grey60 module. *AAT* (LOC103436790, LOC103447337, LOC103453624) is positively correlated with the non-lipoxygenase pathway naringin dihydrochalcone (0.813) and negatively correlated with alpha-linolenic acid (-0.808) in the bisque4 module. In the carotenoid pathway, *CCD* (LOC103438403) and trans-beta-ionone (0.958) are positively correlated in the dark slate blue module. *VDE* (LOC103414012) is positively correlated with lutein (0.901) and neoxanthin (0.829) in the light cyan module (Appendix A).

### 3.4. Real-Time Fluorescence Quantitative PCR (RT-qPCR) Analysis

In order to further confirm the reliability of the data, qPCR was performed to validate the nine major hub genes, and the relative expression levels of peel and pulp tissues collected from Ningqiu apples at three reproductive periods (S1, S2, S3) were analyzed using qRT-PCR; the results show that the trends of transcriptome sequencing (FPKM) and qPCR results are basically the same (Figure 7). Correlation analysis also showed a high degree of consistency between the transcriptome sequencing data and the qPCR results, with correlation coefficients of 0.8 and 0.9 (Appendix A), respectively, indicating that the data are reliable.

## 4. Discussion

Apple ripening is a complex process characterized by fine transcriptional regulation that triggers a profound remodeling of metabolic compound production, which has a significant impact on the aroma of “Ningqiu” apples [45]. In this study, we comprehensively investigated the transcriptomic and metabolomic changes that occur during the ripening of Ningqiu apple peel and pulp and identified the central genes that may play a role in the metabolism of apple aroma compounds.

### 4.1. Terpenoid-Pathway-Related Genes Are Promising Candidates for Understanding the Aromatic Properties of Ningqiu Apples

During the maturation process, the accumulation of terpenoids and the activity of biosynthesis-related genes reveal the tissue specificity of Ningqiu apples. In particular, monoterpene precursors often reach their peak in the skin. At the level of metabolomics and transcriptomics, our results showed that the content of terpene precursors was higher in the skin during color transition and maturation and exhibited more diverse biochemical phenotypes in early terpene VOCs, which confirmed the previous findings of Yan D [46] and Espino-Díaz M et al. [47]. From a genetic perspective, our data indicate that the terpene synthase gene is in a leading position in the skin. As is well known, *TPS* is a key enzyme in terpene biosynthesis, and the functions of several *TPS*-encoding genes have been elucidated. For example, Zhang et al. [48] found that *TPSs* are mainly responsible for the development of fruit pulp and peel, including geraniol, farnesene, and geraniol. However, Pechus SW et al. confirmed the positive role of TPSs in the biosynthesis of (E, E)-α-farnesene and its derivatives via cloning and the functional expression of (E, E)-α- farnesene synthase cDNA from apple fruit periarp tissues [49]. In our study, we found a strong correlation between the *TPS* (LOC103435869) gene and the synthesis of terpenoids such as geranial and (E, E)-α-farnesene in fruit peel tissue. Our gene metabolites revealed a novel association between two candidate compounds (santolina triene and (−)-β-chamigene) which has not been described in the literature. Considering that the biochemical function of enzymes cannot be predicted solely based on sequence similarity, the inconsistency between our correlation and TPS annotation is not surprising. In fact, changes in just a few amino acids can alter the catalytic mechanism of enzymes or lead to completely different product profiles. In red Sichuan pepper, *FDPS2* and *FDPS3* have the highest correlation with limonene, the terpenoid compound with the highest content, indicating that they may be key genes for terpenoid synthesis [50]. We also obtained similar results in our study, where *FFDPS1.X1* (LOC103445233) showed a strong correlation with the synthesis of germacrone (isoprene) in both the skin and pulp. Therefore, we believe that *TPS7* (LOC103435869) and *FDPS1.X1* (LOC103445233) identified by our analysis are promising candidates for the subsequent functional characterization of terpenoids. From a practical perspective, identifying key genes and metabolites responsible for terpenoid VOC and precursor synthesis during apple growth is very interesting for apple cultivation and deep processing, as it can improve decision-making in the production chain.

### 4.2. Fatty Acid Metabolism Aroma Characterization of Promising Candidate Genes for Understanding the Aromatic Properties of Ningqiu Apples

The composition and concentration of aroma compounds are significantly influenced by varietal characteristics and the developmental stages of the fruit [51]. As ripening progresses, the profile of volatile compounds undergoes dynamic changes. Our findings reveal that aldehydes dominate the volatile compound profile during the early stages of fruit development, but their levels decline as the fruit matures, accompanied by a notable increase in alcohol content. Ultimately, esters become the predominant volatile compounds during the ripening phase. The biosynthesis of these compounds is regulated by key genes involved in fatty acid metabolism, including *LOX, HPL, ADH*, and *AAT*, which play pivotal roles in the formation of aldehydes, alcohols, and esters [52,53,54]. Lipoxygenase (*LOX*) initiates the biosynthetic pathway of these compounds by catalyzing the degradation of fatty acids, with linoleic acid and linolenic acid serving as primary substrates [55]. Alcohol dehydrogenase (*ADH*) facilitates the reduction of aldehydes to alcohols, although its transcriptional activity diminishes as the fruit develops [56]. For instance, in “Red Chief Delicious” apples (Malus domestica Borkh), *ADH*-mediated reactions are critical for the biosynthesis of 1-butanol, 2-methyl-1-butanol, and 1-hexanol in various fruit tissues [57]. In the mature fruit of the “Ruixue” variety, aldehydes such as nonanal, decanal, hexanal, 2-hexenal, octanal, and (E)-2-octenal are the predominant aroma compounds, with their expression regulated by ADH through the fatty acid metabolism pathway [58]. Previous studies have identified hexanal and 2,4-decadienal as the primary oxidation products of linoleic acid, while linolenic acid oxidation yields 2,4-heptadienal. These aldehydes are further metabolized by *ADH* to produce additional volatile compounds [59]. In our research, we observed strong correlations between specific *ADH* isoforms and key aroma compounds. For example, *ADH2* (LOC103432769) in fruit peel tissue is highly associated with 1-butanol and 1-hexanol, while *ADH3* (LOC103439766) in the pulp shows a strong correlation with (E)-2-octenal and hexanal. Similarly, *ADH9* (LOC103418393) is linked to nonanal, decanal, and octanal, while *ADH4* (LOC114827344) is associated with (E)-2-octenal and hexanal. Additionally, *ADH2* (LOC103432769) exhibits a strong relationship with β-cyclocitral, (E)-2-heptenal, and 2,4-decadienal. These findings suggest that functional diversification has occurred within the *ADH* gene family during evolution, with individual members potentially specializing in the synthesis of distinct alcohols.

*HPL* functions by breaking down fatty acid hydroperoxides, such as 13-hydroperoxide linolenic acid or 13-hydroperoxide linoleic acid, into short-chain volatile aldehydes (e.g., hexanal and (E)-2-hexenal) and ketones [60,61,62]. Our results corroborate these findings, revealing a strong correlation between *HPL2* (LOC114819168) and (E)-2-heptenal in the pulp tissue. *AAT*, on the other hand, catalyzes the formation of ester compounds (e.g., hexyl acetate and butyl acetate) through acyl transfer reactions between alcohols and acyl-CoA, which are essential for apples’ aroma [63]. *AAT* activity increases progressively with fruit ripening, reaching its maximum in mature fruits [64]. In a comparative study of apple cultivars, approximately one-third (31/102) of the 102 analyzed varieties exhibited the specific MdAAT1 haplotype H1 (C-A-C-A), which was characterized by a marked reduction in ester concentrations. In contrast, the contrasting haplotype H8 (T-G-T-G) was identified in 28 cultivars but was associated with normal or elevated ester levels. The observed associations suggest a putative causal functional relationship between MdAAT1 and the biosynthesis of key apple esters [65]. In our study, *AAT3* (LOC103436790) and *AAT1* (LOC103447337) in the peel tissue were strongly associated with esters such as butanoic acid hexyl ester, butanoic acid butyl ester, L-alanine methyl ester, 2-methyl butanoic acid butyl ester, butanoic acid pentyl ester, 2-methyl butanoic acid heptyl ester, 2-methyl propanoic acid hexyl ester, propanoic acid hexyl ester, and 2-methyl butanoic acid hexyl ester. In the pulp tissue, *AAT3* (LOC103436790) and *AAT1* (LOC103447337) were highly correlated with naringin dihydrochalcone and alpha-linolenic acid. Overall, the combined analysis of aroma compounds, including esters, alcohols, aldehydes, and terpenes, underscores the central role of core hub genes *AAT1* (LOC103447337), *AAT3* (LOC103436790), and *ADH2* (LOC103432769) in the fatty acid pathway, with their coordinated activity driving aroma biosynthesis in both peel and pulp tissues.

### 4.3. Carotenoid Metabolism Aroma Characterization of Promising Candidate Genes for Understanding the Aromatic Properties of Ningqiu Apples

We also explored the carotenoid metabolic pathway, as carotenoids and their derivatives, norisoprenoids, play a significant role in apple fruit aroma, contributing floral and fruity notes to apple by-products [66]. In ripe apples, the predominant carotenoids are β-carotene and lutein, which constitute nearly 85% of the total carotenoid content [67]. However, in our study, the accumulation of these compounds was minimal, and the expression of *CCD1* and *CCD2* genes was down-regulated. This may be attributed to the developmental stage of the fruit, as β-carotene and lutein are typically abundant during the pea size stage but degrade as the fruit matures [68]. Carotenoids serve as precursors for norisoprenoid compounds. Due to their highly conjugated double-bond structures, carotenoids are inherently unstable and can undergo chemical or enzymatic degradation to form norisoprenoids containing carbonyl groups. For instance, α- and β-carotenoids can be cleaved into sesquiterpene ketones such as α-ionone and β-ionone. *CCD3*, in particular, catalyzes the cleavage of β-carotene at the 9,10 (9′,10′) position to generate β-ionone [69]. Our findings align with this, as *CCD3* (LOC103438403) exhibited a significant positive correlation with trans-β-ionone in both peel and pulp tissues. Both trans-β-ionone and β-ionone are produced through the carotenoid cleavage pathway, with *CCD* enzymes facilitating the oxidative cleavage of β-carotene to yield β-ionone and its isomers, such as trans-β-ionone. Notably, trans-β-ionone may impart a stronger or more refined floral aroma compared to β-ionone.

## 5. Conclusions

Deciphering specific patterns of volatile compounds in apples is critical for understanding their biosynthesis during fruit ripening, which can help to characterize high-value single-variety fruits or optimize their processing. In this study, we analyzed key metabolomic and transcriptomic data from the peel and pulp of Ningqiu apples. Among the nearly 108 aroma-related differentially expressed genes (DEGs) identified in both tissues, only 10% were highly correlated with each other, making them ideal candidates for functional studies to validate their role in aroma production. The presence of esters, aldehydes, alcohols, and terpenoids in both the peel and pulp, as well as key genes involved in fatty acid, carotenoid, and terpenoid metabolic pathways, underscores their importance in shaping the variety’s aroma profile. The metabolic markers and candidate genes identified in this study will provide the basis for future functional studies, particularly through gene editing approaches.

## Figures and Tables

**Figure 1 plants-14-01165-f001:**
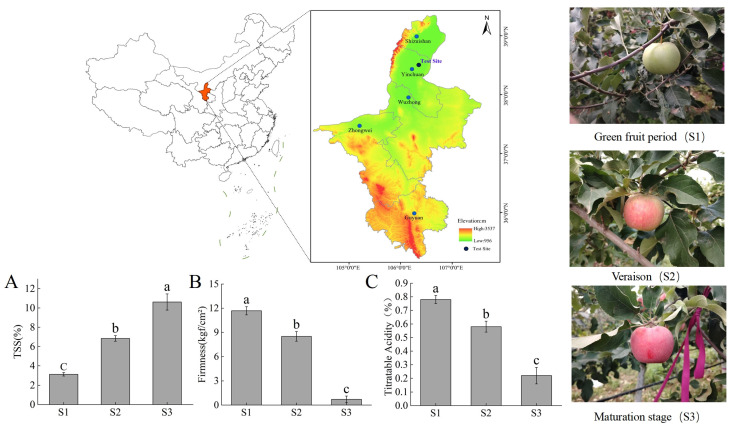
Physiological characteristics of “Ningqiu” apples during ripening. (**A**) Soluble solids, (**B**) fruit hardness, (**C**) titratable acid content. Letters indicate significant differences according to Tukey’s test, *p* < 0.05. Error lines indicate standard error (SE).

**Figure 2 plants-14-01165-f002:**
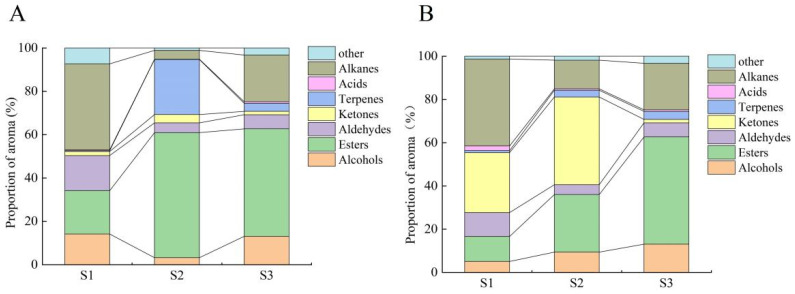
The dynamic characteristics of the proportions of aroma determinants during the ripening process of apples. (**A**) Peel, (**B**) pulp.

**Figure 3 plants-14-01165-f003:**
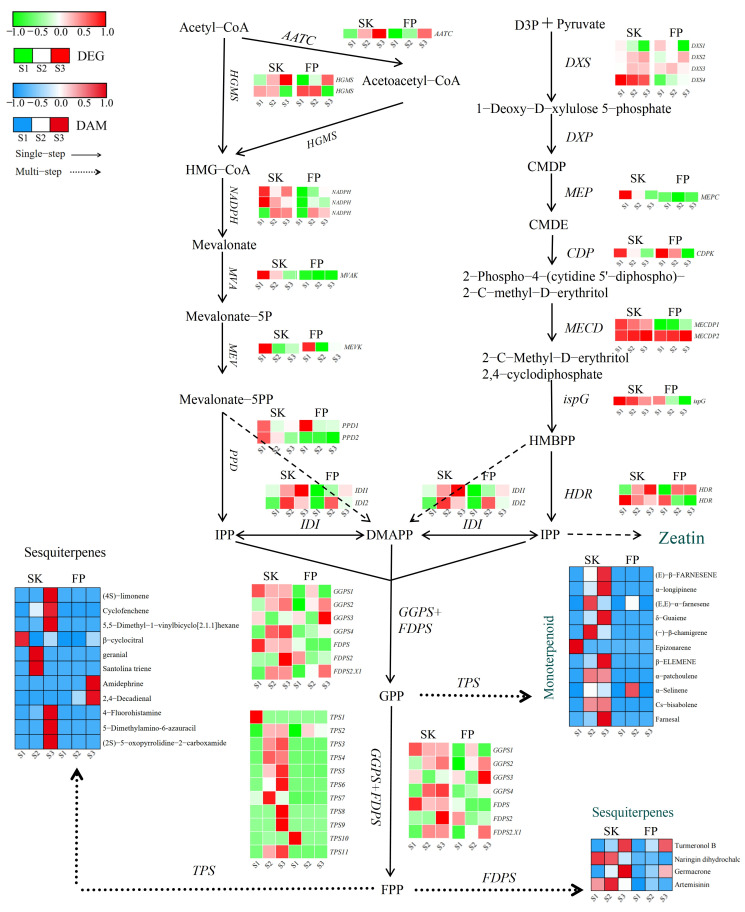
Terpene biosynthesis pathways. Expression levels of DEGs are reported in red–green scaled boxes, while the accumulation of DAMs is reported in blue–red scaled boxes. For each transcript/metabolite, the abundance level is represented by a heat map, where the blocks shown from left to right represent the three stages of the peel and pulp: green fruiting stage (S1), color change stage (S2), and ripening stage (S3). Acetyl-CoA: acetyl coenzyme A; Acetoacetyl-CoA: acetoacetyl coenzyme A; HMG-CoA: hydroxymethylglutaroyl coenzyme A; Mevalonate: 3,5-dihydroxy-3-methylvaleric acid; Mevalonate-5P: (R)-mevalonic acid 5-phosphate; Mevalonate-5PP: (R)-5-diphosphomevalonic acid; IPP: isopentenyl diphosphate; DMAPP: dimethylallyl diphosphate; D3P + Pyruvate: D-glyceraldehyde 3-phosphate + pyruvic acid; 1-Deoxy-D-xylulose 5-phosphate: 1-deoxy-D-xylulose 5-phosphate; CMDP: 2-C-methyl-D-erythritol 4-phosphate; CMDE: 4-(cytidine 5′-diphospho)-2-C-methyl-D-erythritol; 2-phospho-4-(cytidine 5′-diphospho)-2-C-methyl-D-erythritol; 2-C-methyl-D-erythritol 2,4-cyclodiphosphate; HMBPP: 1-hydroxy-2-methyl-2-butenyl 4-diphosphate; GPP: geranyl diphosphate; FPP: farnesyl pyrophosphate; GGPP: geranylgeranyl pyrophosphate.

**Figure 4 plants-14-01165-f004:**
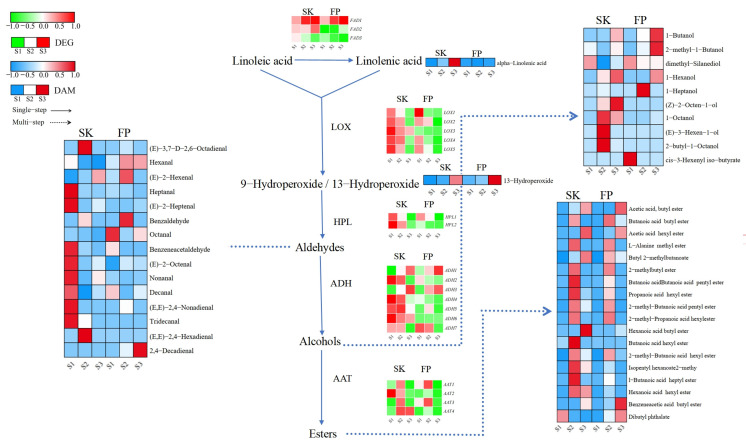
Fatty acid metabolism pathway. Expression levels of DEGs are reported in red–green scaled boxes, while the accumulation of DAMs is reported in blue–red scaled boxes. For each transcript/metabolite, the abundance level is represented by a heat map, where the blocks shown from left to right represent the three stages of the peel and pulp: green fruiting (S1), color change (S2), and ripening (S3).

**Figure 5 plants-14-01165-f005:**
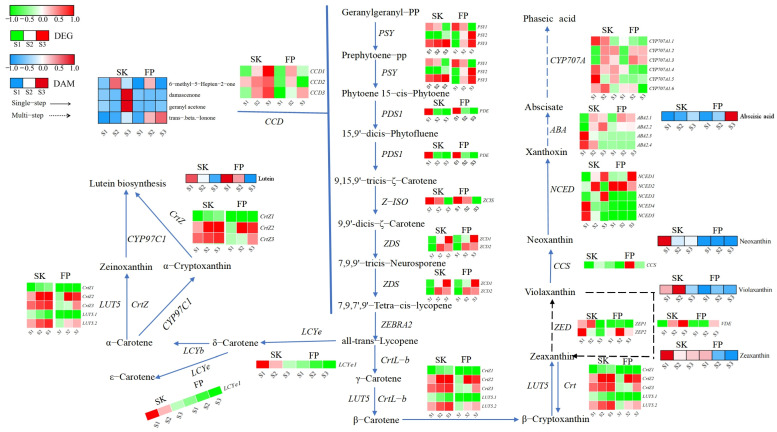
Carotenoid metabolic pathway. Expression levels of DEGs are reported in red–green scaled boxes, while the accumulation of DAMs is reported in blue–red scaled boxes. For each transcript/metabolite, the abundance level is represented by a heat map, where the blocks shown from left to right represent the three stages of the peel and pulp: green fruiting (S1), color change (S2), and ripening (S3).

**Figure 6 plants-14-01165-f006:**
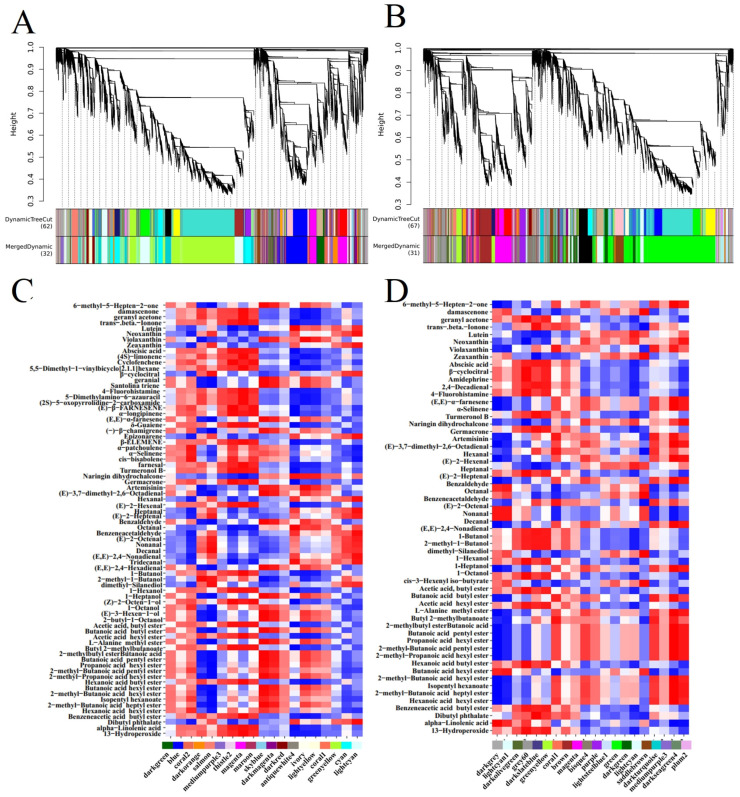
WGCNA of the correlation of highly co-expressed modules (gene cluster color blocks) and identified modules with the content of volatile metabolites and their precursors in the peel (**A**) and pulp (**B**). Red and blue colors indicate positive and negative correlations with gene expression, respectively. (**C**) represents the heatmap analysis of correlations between modules containing at least one previously described aroma-related gene in the fruit peel and metabolites. (**D**) represents the heatmap analysis of correlations between modules containing at least one previously described aroma-related gene in the fruit pulp and metabolites.

**Figure 7 plants-14-01165-f007:**
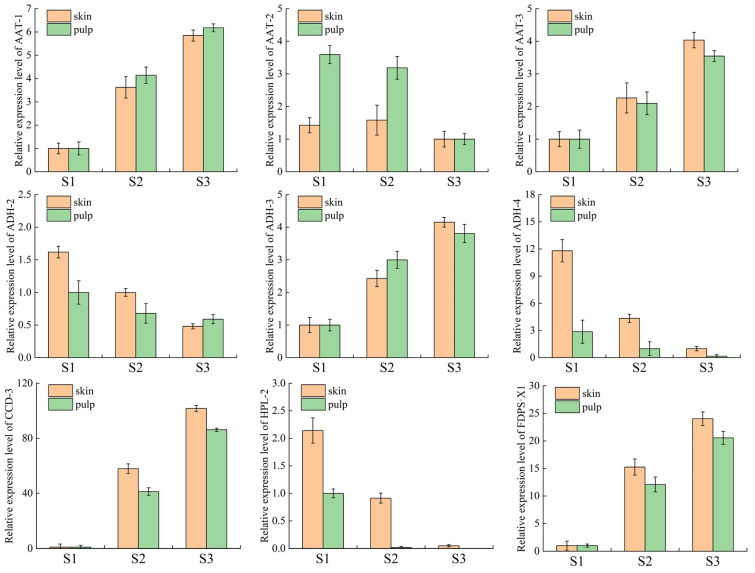
Validation of qRT-PCR for genes related to volatile compound synthesis at three ripening stages in the peel and pulp.

## Data Availability

All data that support the findings of this study are included in this manuscript and its Appendix A.

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
