# Peer review of "Integrated Analysis of the Metabolome and Transcriptome During Apple Ripening to Highlight Aroma Determinants in Ningqiu Apples"

_plants, 2025, doi:10.3390/plants14081165_

Round 1
Reviewer 1 Report
Comments and Suggestions for Authors
The submitted manuscript (plants-3544989) highlights the significant variations in the types and concentrations of volatile aroma compounds and the expression of associated genes in the peel and pulp of apple germplasm during ripening. The authors performed good experimental analysis and got satisfactory results; however, the written manuscript needs minor revision for further refinement. Here are my comments
-The English language needs minor editing for refinement.
-Gene names should be written in italic style. I
-The complete names of all abbreviations should be written the first time writing, and then you can add the short abbreviations in the entire text.
Abstract:
-Gene IDs should be mentioned along with their annotation.
-Line 13: The information about the adopted methodology is missing. Please elaborate in 2-3 lines.
-Line 23: Add the scope and application of the current findings.
Introduction:
-Line 78: There is a need to add the paragraph of literature on the importance of the application of transcriptomic and metabolomic analysis of different fruits.
Methods:
-Please add the detailed plant and fruit characteristics of each variety (Good Delicious × Red Astrachan).
-Please mention the citations and references for each measurement of fruit quality traits, as well as for transcriptomic and metabolomic analysis.
-The raw data reads should be submitted on the NCBI website and provide the PRJN detail for the access feasibility of other researchers.
-Line 265: What was the name of the internal control gene? Please add the full, complete details of all exported gene primers in the supplementary table.
Results:
-Figure 1: Please add the consistent font style and size for statistical letters in Figure 1 and also in all relevant figures of the manuscript.
-What does mean by firmness (gf/cm2)? Please correct the unit in the entire text.
-Discussion: It is well written but still needs to add more literature on comparative studies results about identified genes modulating the fruit quality traits in apples and other horticultural fruits.
References: These should be written in the proper format of the journal.
Comments on the Quality of English LanguageThe English could be improved in minor form to more clearly express the research.
Author Response
Please refer to the attachment for specific responses(PDF).

Reviewer 2 Report
Comments and Suggestions for Authors
The work does not have a clearly defined goal. The authors provide a method for determining fruit quality in the material and methods Line 104-109, which is then described in the results Line 274-287. However, they do not address this at all in the conclusions. Therefore, I propose to move the entire part included in the results (Line 274-287) to the material and methods chapter (because this is the described material used in the experiment).
Line 20, 295, 303, 535, 548, 554: The authors mistakenly use the term pericarp as peel. So please replace pericarp with peel (as it is called in the Material and Methods section Line: 114 Pericarp: The fruit wall, derived from the ovary wall and consisting of up to three layers: exocarp, mesocarp, and endocarp.Peaches are a type of fruit called a drupe. In drupes, the pericarp is made up of the endocarp (outer shell of the pit), mesocarp (juicy flesh), and exocarp (outer skin). Apples are a type of fruit called a pome. In pomes, the pericarp is just the core of the fruit, made up of the endocarp (tough, thin layer surrounding the seeds), mesocarp (the flesh of the core), and the endocarp (the outer layer of the core, which is fused to the edible flesh of the apple). The part of the apple we eat is not derived from the ovary at all, but is an enlargement of the flower receptacle. (https://torontobotanicalgarden.ca/blog/word-of-the-week/botanical-nerd-word-pericarp/)
Line 71, 72, 73The authors should replace the word "variety" with "cultivar". The difference between cultivar vs variety is human involvement. A cultivar is purposely created by humans to enhance traits chosen through artificial selection. A variety is a version of the plant species that occurs naturally through natural selection.
Line 79-88: this is not a description of the goal of the experiment, but rather the results/conclusions. Please state the goal of the experiment correctly.
Line 281: in scientific language, terms such as "generally speaking" are avoided;
Line 291, 329, 331, 334 – will the “Supplemental Material” be visible after publication and in what form? If not, there is no need to cite it in the text,
Line 554-apple is not a berry
Line 666 – scientific language should be precise. Please decide on the word "pulp" or "flesh" and use one term consistently.
Comments on the Quality of English LanguageLine 20, 295, 303, 535, 548, 554: The authors mistakenly use the term pericarp as peel. So please replace pericarp with peel (as it is called in the Material and Methods section Line: 114
Line 71, 72, 73The authors should replace the word "variety" with "cultivar". The difference between cultivar vs variety is human involvement. A cultivar is purposely created by humans to enhance traits chosen through artificial selection. A variety is a version of the plant species that occurs naturally through natural selection.
Line 281: in scientific language, terms such as "generally speaking" are avoided;
Line 554-apple is not a berry
Line 666 – scientific language should be precise. Please decide on the word "pulp" or "flesh" and use one term consistently.
Author Response

(The authors gave the same response as above.)

Round 2
Reviewer 1 Report
Comments and Suggestions for Authors
In my opinion, the authors revised the manuscript in satisfactory form. So, it is accepted in its present form.
Author Response
非常感谢您审阅我们的手稿。我们现在根据审稿人的意见修改了手稿。大多数修订都包含在手稿中。关于我们手稿的修订有几种解释(见附件)。

Reviewer 2 Report
Comments and Suggestions for Authors
The authors responded to all my comments. The work has also been improved from a linguistic point of view, so in my opinion it is suitable for printing.
Author Response
Thank you very much for reviewing our manuscript. We have now revised the manuscript based on the comments of the reviewers. Most of the revisions are included in the manuscript. There are several explanations regarding the revision of our manuscript (see attachment).
